

# Predictions of satellite retrieval failures of air quality using machine learning

Edward Malina[1,2], Jure Brence[3,4], Jennifer Adams[5], Jovan Tanevski[3,6], Sašo Džeroski[3,4,7], Valentin Kantchev[8], and Kevin W Bowman[1]

[1]Jet Propulsion Laboratory, California Institute of Technology, Pasadena, California, USA
[2]Currently at ESA/ESRIN, Frascati, Italy
[3]Jozef Stefan Institute, Ljubljana, Slovenia
[4]Jozef Stefan International Postgraduate School, Ljubljana, Slovenia
[5]University of Zurich, Department of Geography, Zurich, Switzerland
[6]Heidelberg University and Heidelberg University Hospital, Heidelberg, Germany
[7]Φ-lab, ESA/ESRIN, Frascati, Italy
[8]Swift Software Group, Glendale, California, USA

**Correspondence:** Edward Malina (edward.malina.13@ucl.ac.uk)

**Abstract.** The growing fleet of Earth Observation (EO) satellites is capturing unprecedented quantities of information about the concentration and distribution of trace gases in the Earth's atmosphere. Depending on the instrument and algorithm, the yield of good remote soundings can be a few percent owing to interferences such as clouds, non-linearities in the retrieval algorithm, and systematic errors in radiative transfer algorithm leading to inefficient use of computational resources. In this study, we investigate Machine Learning (ML) techniques to predict failures in the trace gas retrieval process based upon the input satellite radiances alone allowing for efficient production of good-quality data. We apply this technique to ozone and other retrievals using measurements from two sets of measurements: Suomi National Polar-Orbiting Partnership Cross-Track Infrared Sounder (Suomi NPP CrIS), and joint retrievals from Atmospheric Infrared Sounder (AIRS) - Ozone Monitoring Instrument (OMI). Retrievals are performed using the MUlti-SpEctra, MUlti-SpEcies, Multi-SEnsors (MUSES) algorithm. With this tool, we can identify 80% of ozone retrieval failures using the MUSES algorithm, at a cost of 20% false positives from CrIS. For AIRS-OMI, 98% of ozone retrieval failures are identified, at a cost of 2% false positives. The ML tool is simple to generate and takes <0.1 s to assess each measured spectrum. The results suggest this tool can be applied to many EO satellites, and reduce the processing load for current and future instruments.

## 1 Introduction

The advent of geostationary Earth Observation (EO) satellites designed to provide hourly estimates of trace gas concentrations is a significant step forward in understanding global problems such as climate change and air pollution (NASES, 2018; Szopa et al., 2021). These satellites, such as Sentinel-4 on MetOp (Ingmann et al., 2012); Troposheric emissions: Monitoring of pollution (TEMPO) (Zoogman et al., 2017); the Geostationary Environmental Monitoring Spectrometer (GEMS) (Nicks et al., 2018), Geostationary Carbon Cycle Observatory (GeoCarb) (Moore III et al., 2018); AIM-NORTH (Nassar et al., 2019) and



Geostationary and Extended Orbits (GeoXO;(NOAA)) are expected to capture many-times more measurements than current Low Earth Orbiting (LEO) instruments, with GeoCarb alone expected to provide hourly estimates of pollutants over North America (Moore III et al., 2018). LEO satellites, such as the Suomi National Polar-Orbiting Partnership Cross-Track Infrared Sounder (Suomi NPP CrIS) captures millions of measurements daily. This results in a significant challenge, i.e., the timeliness of generating trace gas concentrations. The retrieval algorithms required to convert measured spectra into trace gas concen-

trations (Rodgers, 2000; Worden et al., 2007) are resource intensive, typically requiring several minutes to generate a single estimate. Therefore, one of the key challenges in exploiting the capabilities of the geostationary EO satellites is not in making the measurements but in the ability to process, store, and interpret the satellite measurements in a timely manner. This is recognised in the satellite retrieval community, with the TROPOMI total column ozone retrieval algorithm consisting of two aspects, a 'Near Real Time (NRT)' and an 'Offline' version. Where the NRT sacrifices accuracy for speed (Garane et al., 2019).

There has been significant effort dedicated to improving the speed of retrieval algorithms (Hedelt et al., 2019; Noël et al., 2022). The largest bottleneck is found in the Radiative Transfer Models (RTMs) e.g. Vector LINearized Discrete Ordinate Radiative Transfer (VLIDORT;Spurr (2006)). RTMs simulate the transfer of radiation through the atmosphere and are fundamental components of any retrieval algorithm (Rodgers, 2000). Typical speed-up methods include replacing the whole or part of the RTM with an approximation such as an emulators/Neural Network (NN) (e.g. Rivera et al. (2015); Brence et al. (2023);

Brodrick et al. (2021); Bue et al. (2019); Efremenko et al. (2014a, b); Himes et al. (2020); Pal et al. (2019)) or look-up-table (Loyola et al., 2020). Other methods include simplifying the input and output of the RTM using techniques such as Principle Components Analysis (PCA) (Jindal et al., 2016) or reducing the number of monochromatic calculations (Efremenko et al., 2014a, b; Kopparla et al., 2016, 2017; Liu et al., 2020; Mauceri et al., 2022; Natraj et al., 2005, 2010; Somkuti et al., 2017; Spurr et al., 2013).

However, a significant drain on resources while processing large quantities of spectra still remains; the fact that the retrieval process frequently yields poor quality results, where the output data must be discarded. These retrieval failures can occur for a variety of reasons, for example, excessive cloud in the light path, low Signal-to-Noise Ratio (SNR), poor quality fit, or many other reasons depending on the algorithm in question (Kulawik et al., 2021). These failed retrievals require the same processing resources as good quality retrievals; if those spectra that yield failed retrievals could be screened and removed from

the processing chain, then significant processing overhead could be saved.

In this study, we investigate Machine Learning (ML) methods for predicting failed trace gas retrievals using measured satellite spectra prior to full retrieval. Some research has been conducted on the pre-selecting or filtering of trace gas retrievals using ML methods based on genetic algorithms (Mandrake et al., 2013). While other examples exist where NNs are used to improve the throughput from retrieval algorithms (Mendonca et al., 2021). However, in the case of Mendonca et al. (2021), the

method is applied purely over northern latitudes, for a specific solution not applicable to most satellite instruments, while the solution presented in this paper will be applicable to any satellite instrument on a global basis.

The primary source of data for this study are spectra from the satellite instruments Suomi NPP CrIS (Han et al., 2013), Atmospheric Infrared Sounder (AIRS) on the Aqua satellite (Aumann et al., 2003), and the Ozone Monitoring Instrument (OMI) on Aura. Trace gas retrievals and their associated quality statistics are generated using the MUlti-SpEctra, MUlti-



SpEcies, Multi-SEnsors (MUSES) retrieval algorithm (Worden et al., 2007; Luo et al., 2013; Fu et al., 2013, 2016, 2018; Malina et al., 2024), which is a core part of the TRopospheric Ozone and its Precursors from Earth System Sounding (TROPESS) project. TROPESS produces long-term, Earth science data records with uncertainties and observation operators, freely available (https://tes.jpl.nasa.gov/tropess/get-data/products/). The MUSES algorithm has considerable heritage for instruments sensitive to a wide range of spectral regions, from Ultra Violet (UV) to Thermal Infrared (TIR) (Bowman et al., 2002; Kulawik et al.,

2006; Malina et al., 2024; Worden et al., 2007; Natraj et al., 2011, 2022; Luo et al., 2013; Fu et al., 2013, 2018; Kulawik et al., 2021). The CrIS instrument was chosen for this analysis due to the high data volume, long spectral range, allowing for multiple different products, and current use of CrIS products in TROPESS. For example, CrIS ozone retrievals have been used with reanalysis models to understand tropospheric ozone during COVID-19 lockdowns (Miyazaki et al., 2021). The carbon monoxide products of TROPESS / CrIS have been used to understand the impact of wildfires in Australia (Byrne et al., 2021).

The joint spectral products of TROPESS from AIRS-OMI (Fu et al., 2013) were also chosen for this study, due to the inclusion of the OMI UV sensitivity, which contrasts to the TIR of CrIS, and the application of this product (Miyazaki et al., 2019).

    Different trace gas (e.g. ozone or carbon monoxide) retrievals absorb in different spectral windows, meaning that each gas retrieval has different characteristics and will not fail in the same way. Therefore we focus on three different MUSES CrIS and AIRS-OMI products in this study, namely ozone ($O_3$), carbon monoxide (CO), and temperature profile (TATM) to explore

these differences. These products were chosen for sensitivities in different regions of the CrIS spectral range. Further, although this study is focused on the MUSES algorithm and data from the CrIS and AIRS-OMI instruments, the methods are readily applicable to any retrieval algorithm or satellite instrument.

    This paper is structured as follows: Sect. 2 describes the satellite data and atmospheric retrieval methods used in this study. Section 3 identifies the training datasets that form the core of the study, and the ML tools that use them. Section 4 shows the

performance of the ML models, and Sect. 5 applies the ML models to a dataset not seen during training. The discussion and conclusion are presented in Sects. 6 and conclusions.

## 2   Instruments and tools

### 2.1   Suomi NPP CrIS

CrIS is a nadir viewing Fourier Transform Spectrometer (FTS) that measures TIR radiances in three spectral bands identified

in Table 1 (Han et al., 2013). Located on the Suomi NPP satellite (operational since 28 October 2011), in a near-polar, sun-synchronous, 828 km altitude orbit with a 13:30 ascending node crossing time. CrIS provides daily global measurements with a width width of 2300 km, sampled at 30 cross-track positions, each position consisting of a 3x3 array of field of views (pixels) of diameter 14 km. The wide spectral range and high spatial sampling allows CrIS to retrieve a range of atmospheric products, including trace gas products such as ozone and carbon monoxide (Fu et al., 2018; Kulawik et al., 2021; Malina et al., 2024).

With the wide spectral range, multiple trace gas products from the MUSES CrIS algorithm are regularly generated as part of the TROPESS project (https://tes.jpl.nasa.gov/tropess/get-data/products/ (TROPESS)). Offering an opportunity to test the retrieval failure tool on multiple spectral windows from the same instrument.



**Table 1** Characteristics of the Suomi NPP CrIS bands.

| Band (Name) | Spectral range ($cm^{-1}$) | Spectral resolution ($cm^{-1}$)) | Spatial resolution ($km^2$) |
|---|---|---|---|
| **Band 1 (Long-Wave (LW))** | 648.75-1096.25 | 0.625 | 14 (circ) |
| **Band 2 (Mid-Wave (MW))** | 1208.75-1751.25 | 0.625 | 14 (circ) |
| **Band 3 (Short-Wave (SW))** | 2153.75-2551.25 | 0.625 | 14 (circ) |

**Table 2** Characteristics of the Aqua AIRS and AURA OMI bands.

| Band (Name) | Spectral range (nm) | Spectral resolution (nm) | Spatial resolution ($km^2$) |
|---|---|---|---|
| **AIRS (LW)** | 8.8-15.4 | $\lambda/\Delta\lambda 1200$ | 13 (circ) |
| **AIRS (MW)** | 6.20-8.22 | $\lambda/\Delta\lambda 1200$ | 13 (circ) |
| **AIRS (SW))** | 3.74-4.61 | $\lambda/\Delta\lambda 1200$ | 13 (circ) |
| **OMI (UV1))** | 270-314 | 1.0 - 0.45 nm | 13x24 |
| **OMI (UV2))** | 306-380 | 1.0 - 0.45 nm | 13-24 |

## 2.2 AIRS-OMI

AIRS is a grating spectrometer onboard the Aqua satellite, that measures TIR emissions in the 650–2665 $cm^{-1}$ spectral range,
similar to CrIS (Aumann et al., 2003). AIRS is a cross-track scanning instrument that provides daily global coverage of multiple
species with a footprint of ∼13.5 km.

OMI is a nadir-viewing push broom ultraviolet–visible (UV-VIS) grating spectrometer on the AURA satellite that measures
solar back-scattered radiance. OMI measures in the 270–500 nm wavelength range (Levelt et al., 2006). The ground pixel size
of OMI at nadir is ∼13×24 km when using the 310–330 nm spectral range.
TROPESS provides a joint spectral AIRS-OMI ozone product (Fu et al., 2018), combining information from both TIR and
UV. The AIRS-OMI retrieval has been extensively validated, and has been used as a key component for chemical re-analysis
datasets (Miyazaki et al., 2020b, a). The characteristics of AIRS and OMI are identified in Table 2.

## 2.3 TROPESS/MUSES

### 2.3.1 Algorithm description

The MUSES algorithm has a long heritage in retrieving atmospheric parameters and is designed to be flexible, such that
multiple trace gas retrievals from multiple instrument types are possible, including CrIS (CrIS is also on NOAA's Joint Polar
Satellite System (JPSS) NOAA-20), AIRS on Aqua, Tropospheric Emissions Spectrometer (TES),OMI on the AURA satellite
and the TROPOspheric Monitoring Instrument on Sentinel-5P. The description and application of MUSES to these instruments
are described elsewhere (Kulawik et al., 2006; Fu et al., 2013, 2018; Worden et al., 2019; Kulawik et al., 2021; Bowman et al.,
2006; Worden, 2004; Worden et al., 2007, 2012; Malina et al., 2024). However, to summarise, MUSES is a non-linear retrieval
algorithm based on the well established Optimal Estimation Method (OEM) (Rodgers, 2000). The MUSES CrIS retrieval



provides the following retrieval quantities; $O_3$, CO, TATM, $H_2O$, HDO, methane ($CH_4$), ammonia ($NH_3$), peroxyacetyl nitrate (PAN) and methanol ($CH_3OH$). A retrieval pipeline is implemented to refine atmospheric parameters prior to the retrieval of these trace gases. For example, cloud parameters are derived as the first step.

To determine trace gas concentrations, MUSES optimally fits the simulated radiance output from an RTM in predetermined spectral windows to radiance measurements, in the case of CrIS and AIRS, we use the Optimal Spectral Sampling (OSS) RTM (Moncet et al., 2008, 2015), and in the case of OMI, VLIDORT (Spurr, 2006). This allows for the quantification of physical parameters whose variation impacts measured radiance, such as ozone or CO concentrations or atmospheric temperature variations. The OE routine in MUSES computes the best-estimate state vector $\hat{\mathbf{x}}$, which represents the atmospheric state and

ancillary variables, by minimising the following cost function Eq. 1:

$$J(\mathbf{x}) = [\mathbf{y} - \mathbf{F}(\mathbf{x}, \mathbf{b})]^T \mathbf{S}_\epsilon^{-1} [\mathbf{y} - \mathbf{F}(\mathbf{x}, \mathbf{b})] + (\mathbf{x} - \mathbf{x_a})^T \mathbf{S_a}^{-1} (\mathbf{x} - \mathbf{x_a}). \tag{1}$$

The residual between the observed radiance $\mathbf{y}$ and the simulated radiance $\mathbf{F}(\mathbf{x}, \mathbf{b})$ inversely weighted by the instrument error co-variance matrix $\mathbf{S}_\epsilon$. $\mathbf{F}(\bullet)$ represents the forward model, with $\mathbf{b}$ forming a vector of elements necessary for RTM simulation, but not retrieved. The instrument noise $\mathbf{S}_\epsilon$ is obtained from CrIS L1b files. The difference between the retrieval vector $\mathbf{x}$ and

the apriori state $\mathbf{x}_a$ inversely weighted by the apriori covariance matrix $\mathbf{S}_a$ represents the physics element of the OEM. $\mathbf{S}_a$ describes the expected covariance of the apriori state. The state vector parameters for the targets of this research (ozone, CO, and TATM) are indicated in Table A1.

## 3    Machine learning tools and datasets

Predicting retrieval failure is a binary classification task, where the input, in this case a L1b spectrum, contains many continuous

parameters (radiances) and the output is a single binary value, indicating a good or bad quality retrieval. We consider an example positive if its retrieval failed.

### 3.1    Training datasets

Two training datasets for CrIS and AIRS-OMI are employed, each formed of approximately 40000 individual retrievals, obtained over five days (each day contains roughly 8000 points) in the year 2020, with each day taken from a different month to

capture different seasonal effects. We train a separate machine learning model for each MUSES product determined from each instrument: ozone, carbon monoxide, and TATM.

For CrIS training, the first vector to be passed into the ML model is one of two options; the measured spectral data for one of the specified trace gas quantities (i.e., in the spectral windows defined in Tables B1, B3 and B4), or the whole CrIS spectral range. The spectroscopic effects immediately outside the spectral windows can impact the spectral windows of the target gases.

We therefore assess the impact of the whole CrIS spectral range on predicting retrieval failures.



For AIRS-OMI training, only the spectral windows were used as defined in Tables B1, B2, B3 and B4. The whole spectral range was found not to have a significant impact.

The second vector input for training purposes is the quality statistics associated with the L1B spectra. Following the completion of a retrieval, the MUSES algorithm undertakes an assessment of the quality through the flagging of specific metrics. The quality flags for MUSES CrIS/AIRS/AIRS-OMI ozone, carbon monoxide, and TATM retrieval are indicated in Table 4. These values are based on a statistical analysis of the retrieval data indicating the typical ranges. If any of these values are flagged for falling outside of the accepted range, then the retrieval is determined to be of poor quality, tripping a master quality flag. Unique quality values are generated for each target gas and are identical for training purposes whether the spectral window or full band is used.

For training purposes, there are six distinct training datasets for CrIS, with each of these training datasets drawn from the same L1B spectra, the differences being the spectral windows. Therefore, there are two datasets for each of the targets MUSES products (ozone, carbon monoxide, and TATM), with one using the spectral windows defined in Tables B1 or B3 or B4, and one using the full available CrIS spectral range. These are defined in Table 3 and Fig. 1.

For AIRS-OMI there are three training sets based on the spectral windows of the products (ozone, carbon monoxide, and TATM). Note that only AIRS retrieves carbon monoxide and TATM, while the joint AIRS-OMI retrievals is used for ozone.

**Table 3** Description of input training datasets.

| Instrument | Training dataset number | Target Window | Spectral Dimensions | Total failed retrievals |
|---|---|---|---|---|
| CrIS | 1 | Ozone (window only) | 216 | 25% |
| CrIS | 2 | Ozone (Full band) | 2223 | 25% |
| CrIS | 3 | Carbon monoxide (window only) | 31 | 1.7% |
| CrIS | 4 | Carbon monoxide (Full band) | 2223 | 1.7% |
| CrIS | 5 | Temperature profile (window only) | 475 | 27% |
| CrIS | 6 | Temperature profile (Full band) | 2223 | 27% |
| AIRS-OMI | 7 | Ozone (window only) | 369 | 70.6% |
| AIRS | 8 | Carbon monoxide (window only) | 21 | 42.1% |
| AIRS | 9 | TATM (window only) | 442 | 16.6% |



The spectral windows defined in Table 3 for each trace gas are shown in contrast to the available CrIS spectral range in Fig. 1. Both the TATM and ozone spectral regions are largely found in the LW and MW spectral regions, with some overlap between them. Carbon monoxide is only found in a very narrow range in the SW spectral region.

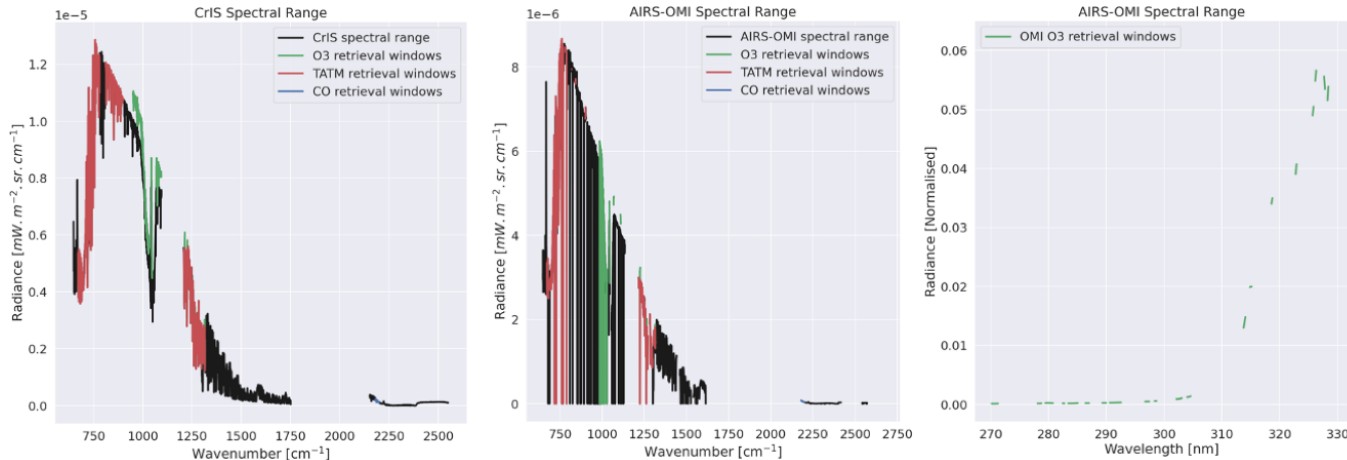

**Figure 1.** Example spectral windows, the left hand panel shows windows for ozone, CO and TATM with respect to CrIS radiance. The Middle panel shows the same as the left hand panel, but for AIRS radiance. The righthand panel shows the ozone spectral windows in the OMI radiance, as a part of the AIRS-OMI retrieval.

Note that numerous gaps are shown in Fig 1 for the OMI radiance values, which are due to poor quality spectral pixels removed from the analysis.



**Table 4** List of quality flags in MUSES for CrIS and AIRS-OMI ozone, carbon monoxide, and temperature profile retrievals, all retrieval values that fall outside the specified range are flagged as bad quality. The pass rate for each instrument for 8000 retrievals on the example day June 15th, 2020 is shown.

| Criteria | Pass rate - CrIS \| AIRS-OMI | Target | Description |
|---|---|---|---|
| Normalised Residual radiance RMSE | 88% \| 79% <br> 91.9% \| 71% | O3 <br> CO | Normalised Residual radiance RMSE |
| Absolute residual radiance mean | 91% \| 90% <br> 92% \| N/A | O3 <br> CO | Absolute residual radiance mean |
| Absolute value of $\mathbf{K} \cdot \Delta \mathbf{L}$ | 84% \| N/A <br> 92% \| 89% <br> 89% \| 95% | O3 <br> CO <br> TATM | The dot product of Jacobian and the residual radiance. |
| Absolute value of $\mathbf{L} \cdot \Delta \mathbf{L}$ | 91% \| N/A <br> 91% \| 95% | O3 <br> CO | Dot product of radiance and residual radiance. |
| Surface temperature - a priori value | 99% \| 99% | O3 | Difference between retrieved surface temperature and initial a priori |
| Cloud top pressure | 91% \| 96% <br> 91% \| 96% <br> 91% \| 98% | O3 <br> CO <br> TATM | Ensures cloud top pressure is within the specified range |
| Mean cloud optical depth | 91% \| 95% <br> 92% \| 96% <br> 92% \| 92% | O3 <br> CO <br> TATM | Ensures cloud optical depth falls within the specified range. |
| Cloud optical depth variability | 91% \| 96% <br> 91% \| 77% <br> 69% \| 85% | O3 <br> CO <br> TATM | Cloud optical depth variation between retrieval windows. |
| Mean emissivity | 92% \| N/A <br> 92% \| N/A | O3 <br> CO | Difference between the retrieved emissivity and the a priori emissivity. |
| Ozone continuum curve | 90% \| 93% | O3 | Checks Ozone slope in the troposphere. |
| Ozone tropospheric consistency | 86% \| 79% | O3 | Compares the initial guess for the tropospheric ozone column and the retrieved column. |
| Ozone column error | 86% \| 96% | O3 | Checks the retrieval error on the ozone column. |
| Cloud Fraction | N/A \| 31% | O3 | Remove cloudy scenes from retrieval, UV only. |



The values shown in Table 4 indicate similar pass thresholds for all of the flags indicated, with some exceptions. For all three targets, the $\mathbf{K} \cdot \Delta \mathbf{L}$ flag generally causes the highest failure rates. Here, $\mathbf{K}$ is the retrieval Jacobian matrix, i.e. a description of the sensitivity of the forward model to changes in the state vector. $\Delta \mathbf{L}$ is the residual radiance after the retrieval, i.e. the difference between the measured instrument radiance and the final simulated RTM radiance. Low values of $\mathbf{K} \cdot \Delta \mathbf{L}$ indicate
that little information remains in the signal, which will occur in challenging retrieval conditions (e.g. high latitudes). TATM retrievals show lower pass rates for cloud optical depth variability, compared to other cloud factors. For ozone only, the lowest pass rate flags are the tropospheric consistency, and most significantly the cloud fraction, which is a factor important only in the UV spectral region. The addition of ozone-specific flags indicates that ozone is a highly challenging gas to retrieve, especially in the troposphere where the dynamics of ozone are still poorly understood (Szopa et al., 2021).

Higher failure rates for ozone and TATM for CrIS shown in Table 3 compared to carbon monoxide can be attributed to the additional flags shown in Table. 4, except in the case of AIRS-OMI ozone, where the majority of failures are due to the cloud fraction flag. The targets described in this paper are retrieved in serial steps using the same spectrum. However, a poor quality retrieval from one target will not necessarily impact the other targets.

### 3.1.1 Training data Resampling and dimensionality reduction

We split the dataset, composed of 40k individual retrievals, into a training set, which contains 80% of samples, and a test set, containing the remaining 20%. In order to avoid biases relating to specific days, we combined the data from all five days, and ensured even distributions in the training and test sets.

The data is moderately imbalanced, since, for example in the case of CrIS, only approximately 25% of ozone examples represent failures, while the opposite is true for AIRS-OMI. Uneven representation of classes often poses problems for classifi-
cation algorithms. A common approach to dealing with unbalanced datasets is to resample the training data so as to simulate a balanced dataset. The simplest methods of balancing the dataset are random undersampling, where we randomly drop a portion of negative (majority class) examples, and random oversampling, where we duplicate a number of copies of positive (minority class) examples, so that the portion of positive examples is close to 50%. We have also considered the more advanced oversampling method Synthetic Minority Oversampling Technique (SMOTE), where synthetic examples are created as a convex
combination of a random positive example and one of its k nearest neighbors (Chawla et al., 2002). Finally, we considered also a combination of oversampling and undersampling, as implemented in SMOTE.

The input for classification are spectral data with variable resolution. We use principal component analysis (PCA) to evaluate whether or not reduced dimensionality of the input spectral data improves the ML pipeline. PCA is a linear method of dimensionality reduction that finds a lower-dimensional representation of the data, so that the explained variance is maximized.

### 3.1.2 The Machine Learning Model

No single machine-learning method is the best choice for every task. Furthermore, different data pre-processing approaches can have a large impact on the ability of models to learn from the data. We refer to a sequence of pre-processing steps and machine learning models as a machine learning pipeline.



In order to identify the most appropriate machine learning pipeline for predicting retrieval failure, we employed the Tree-based Pipeline Optimization Tool (TPOT). TPOT is an automated machine learning method that optimizes machine learning pipelines using genetic programming (Olson et al., 2016). TPOT makes use of the python Scikit-learn library (Pedregosa et al., 2011) and constructs pipelines, composed of the numerous ML tools available (e.g. Neural Networks, Gaussian Processes, etc). For each pipeline TPOT optimizes the hyper-parameters of all its components. TPOT uses internal cross-validation to optimize hyper-parameters and evaluate the performance of each pipeline. We choose the pipeline with the best performance in internal cross-validation as the best pipeline for our task and evaluate its generalization performance on the so-far untouched test set.

The best pipeline for the task of predicting retrieval failures, identified by TPOT, was composed of only one element: extremely randomized trees. We added three pre-processing steps to form the pipeline, shown in Fig. 2.

– 1. Standard scaling. We apply the transformation $\frac{X_i - \mu_{x_i}}{\sigma_{X_i}}$, where i denotes the i-th input dimension (wavelength) and $\mu$ and $\sigma$ represent the respective mean and standard deviation.

– 2. PCA. We perform PCA to reduce the number of dimensions to 30, or the dimensionality of the dataset, whichever is lower. However, note that we have found mixed results when using the PCA transformation. Sometimes the application of PCA improves the predictive performance of the models, and sometimes reduced performance is observed. Therefore in Section 4, we provide results when PCA has been applied, and when it has not.

– 3. Random undersampling. Since the failed retrievals are underrepresented in the data, we balance the dataset by randomly subsampling the majority class. Undersampling is used only during training and is skipped during model evaluation and operational use.

– 4. Extremely randomized trees (Geurts et al., 2006). An ensemble learning technique that constructs a large number of decision tree classifiers – tree-structured models with class labels in leaves and descriptive features in branches. At each branch of a tree, only a restricted subset of features is considered. Both the subset of features and the cut-point choice for each feature are randomized. Samples are classified by a majority vote among the classifications of individual trees. The relative importance of each feature can be estimated by its total contribution to the decrease of class impurity in the nodes of each tree, averaged over the ensemble.

The model takes a L1b spectrum as input and predicts the probability that trace gas retrieval for that spectrum will result in failure. A discriminatory threshold can then be applied to this output probability in order to make a definitive statement on whether or not a pass or fail is predicted. The assumed threshold is 50%, but can easily be changed in the model depending on the requirements of the user.



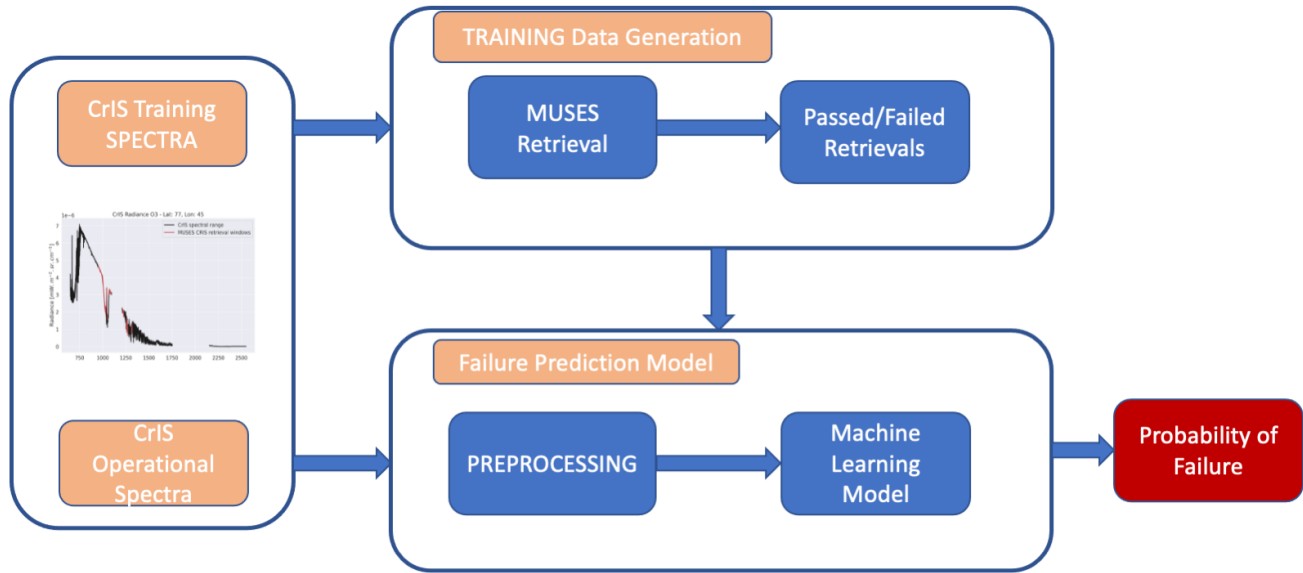

**Figure 2.** Flow chart describing the machine learning pipeline for the failure prediction model.

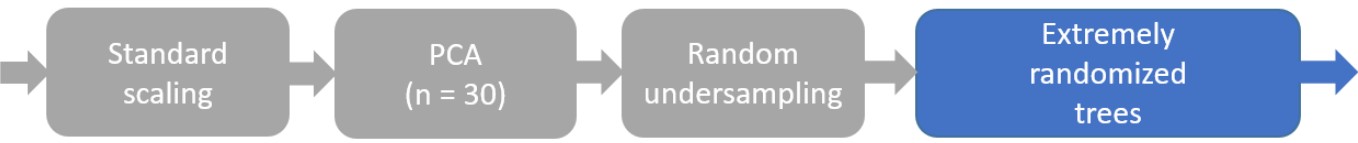

**Figure 3.** Block diagram describing the processing pipeline (the grey boxes) used in the ML model employed in this study, and the ML model chosen by TPOT (the blue box). This figure is an in-depth view of the 'Failure Prediction Model' shown in Fig. 2.

## 4 ML Performance Assessment

An example of how passed and failed retrievals are distributed globally is shown in Fig. 4. We note a number of regions for both TATM and ozone where failure is common, including north and south Africa, as well as parts of China. This figure is for reference; as stated previously, the training and validation datasets are drawn from all five days to avoid bias on a particular day.





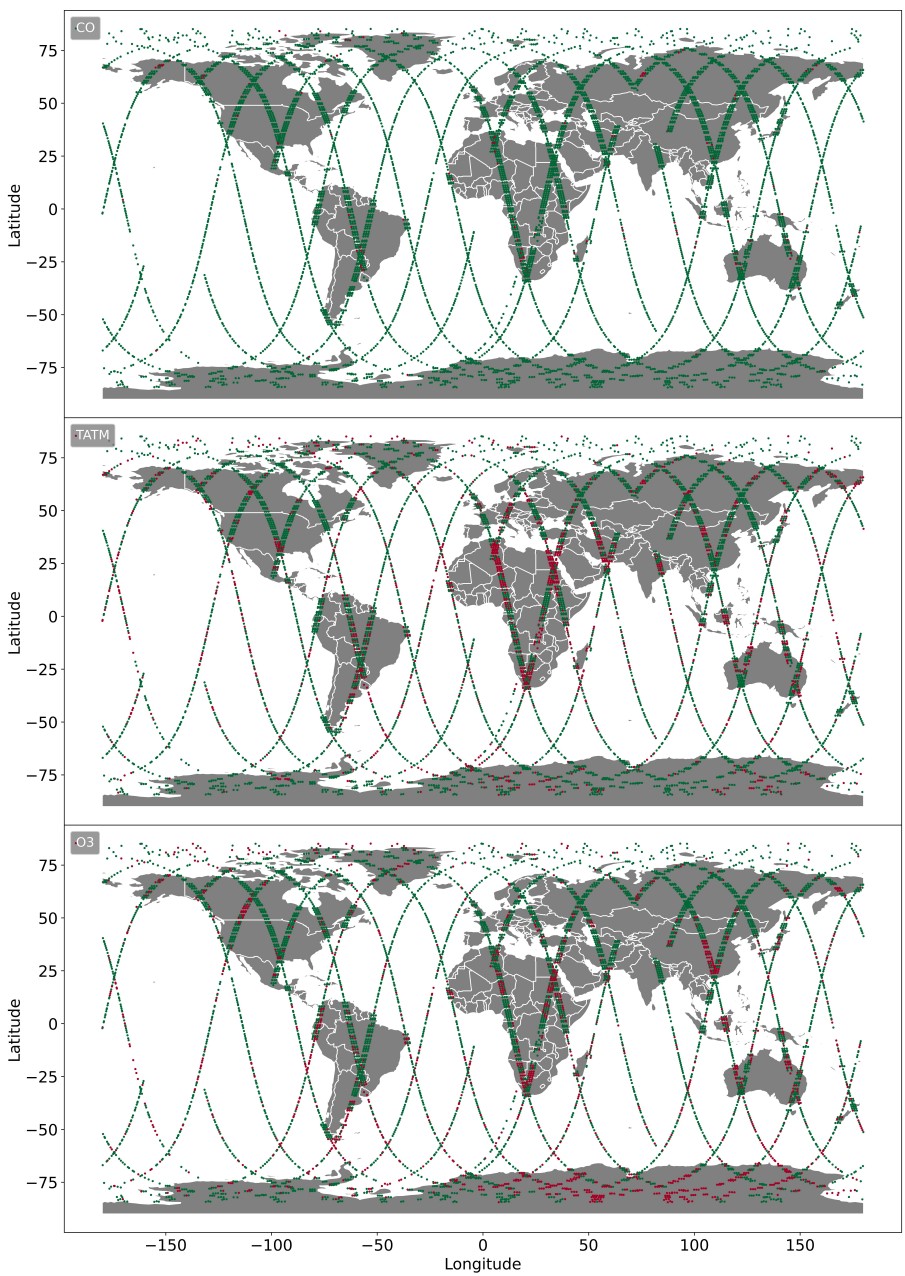

**Figure 4.** Global distributions of failed retrievals on January 15th 2020 for MUSES CrIS carbon monoxide (top), temperature (middle), and ozone (bottom) profile retrievals. Green markers indicate passed retrievals, and red markers show failed retrievals.





## 4.1 Receiver Operator Characteristics (ROC)

For binary classification tasks, common forms of model assessment are ROC curves (Fawcett, 2006). ROC curves compare how many correct positive results are predicted amongst all of the positive samples available (in this case failed retrievals), against

how many incorrect positive results occur amongst all the negative samples (passed retrievals). ROC curves demonstrate the ability of a binary classifier model as its discrimination threshold is varied (as described in Sect. 3.1.2). ROC curves for each of the nine training datasets described in Table 3 are shown to assess the effectiveness of the ML model in each case. The horizontal axis of each graph represents the false positive rate (FPR) $FPR = \frac{FP}{N}$ where FP is the number of false positive predictions and N is the number of all negative examples in the test set. The vertical axis shows the true positive rate (TPR),

with $TPR = \frac{TP}{N}$ where TP is the number of false positive predictions and N is the number of all positive examples in the test set. A perfect model, when represented by an ROC curve, would show all FPR values equal to a TPR of 1.0. The overall performance of the models can be quantitatively described using the Area Under ROC Curve metric (AUC; Flach et al. (2011)). AUC is calculated through numerical integration of the ROC curve and is an effective measure of the probability that a failure will be correctly predicted against the probability that a passed retrieval will be classified as a failure, without committing to

a specific discrimination threshold. An AUC value of 0.5 indicates an uninformed model, with any value above 0.5 showing some benefit to the trained model.





**Figure 5.** ROC curves for the training scenarios 1-6, with the target quantities ozone, carbon monoxide, and TATM (from left to right) from CrIS, input to the ML model has been passed through a PCA. The top row shows ROC curves when the ML tool is trained only on the spectral windows identified in Tables B1, B3, and B4. The bottom row presents ROC curves for when the ML tool is trained on the full CrIS spectral range. The blue lines represent the ROC curve for a specific model, while the black 1:1 line represents an uninformed classifier model. The title for each panel indicates the target trace gas, as well as the AUC score for the target/window case.

Figure. 5 focuses on the results obtained when using PCA in the ML pipeline. The top row of Fig. 5 indicates the clear positive benefit of the models when only using the spectral windows of the relevant trace gases. Both ozone and TATM show superior performance over CO, most likely because of the short spectral window of CO. Both ozone and TATM suggest that
the benefit will come from a low threshold value since high TPR can be achieved with low FPR; for example, for ozone in plot (a), a TPR of 0.5 equals an FPR of 0.1, while a TPR of 0.8 equals an FPR of 0.3. The second row of Fig. 5, indicates the performance of the models when trained on the whole CrIS spectral range. In the case of ozone and TATM, it is not beneficial



to use all of the CrIS spectral range, as indicated by the AUC scores in Fig. 5. Although CO shows a 18% increase in the AUC
score when the entire CrIS spectral range is used, most likely due to the short CO spectral window. The use of PCA in the ML
pipeline is identified in Fig. 6. The whole CrIS spectral range yields the best AUC scores for each of the trace gas cases.

The AUC scores for all PCA and non-PCA cases are shown in Table 5 for comparison purposes. Table 5 shows that the
best results for ozone and TATM are obtained when only training is performed on the relevant spectral windows when PCA is
included in the ML pipeline. For CO, the best results are for the ML pipeline without PCA, and when the ML model is trained
on the whole CrIS spectral band. The results show there is no 'one solution' for the best ML pipeline for trace gas retrieval
failure prediction. This is highlighted by the differences in performance between PCA and non-PCA, with window only Ozone
showing a 8.5% AUC score difference, while the equivalent case for TATM only shows a 2% AUC difference.

**Figure 6.** As Fig. 5, however showing results for the ML model without PCA.





Figure. 7 shows the impact of different instruments and wavelengths on the ML models. The AUC values indicate a substantial improvement in comparison to the CrIS results. For example, the results of the equivalent ozone spectral window show a 4% difference. The CO windows especially show improvement, despite the use of the same spectral window in both instruments. The use of PCA/non-PCA has limited impact on the AUC scores, suggesting the use of PCA in the ML pipeline is not important for AIRS-OMI.

**Figure 7.** ROC curves for the training scenarios 7-8, all Fig aspects are as Fig. 5, however showing results AIRS-OMI ML results for both PCA and non PCA cases.

 

**Table 5** AUC values for the training datasets.

| Training dataset number | Target Window | AUC PCA | AUC no PCA |
|---|---|---|---|
| 1 | CrIS Ozone (window only) | 0.812 | 0.746 |
| 2 | CrIS Ozone (Full band) | 0.788 | 0.786 |
| 3 | CrIS Carbon monoxide (window only) | 0.726 | 0.724 |
| 4 | CrIS Carbon monoxide (Full band) | 0.785 | 0.827 |
| 5 | CrIS Temperature profile (window only) | 0.835 | 0.817 |
| 6 | CrIS Temperature profile (Full band) | 0.799 | 0.822 |
| 7 | AIRS-OMI Ozone (window only) | 0.846 | 0.846 |
| 8 | AIRS CO (window only) | 0.851 | 0.837 |
| 9 | AIRS TATM (window only) | 0.869 | 0.888 |

## 4.2 Feature importance

Section 4.1 shows that spectral data outside of the CrIS spectral windows indicated in Tables B1, B3 and B4 have an influence on the performance of the failure prediction models, especially when PCA is not used. Here we investigate if it is possible to determine which wavelengths have a significant influence on the failure prediction models. The extremely random trees classifier easily provides estimations of the relative importance of features, a measure of the likelihood of misclassification caused by that feature (Geurts et al., 2006; Petković et al., 2020). For cases where PCA is used, the input to the classifier is features, transformed by PCA, meaning that we multiply each feature importance by its PCA loading coefficient and sum over all the principal components, in order to get a feature importance estimate for features in the space of wavelengths. Given the possible impact of the PCA on the performance of the ML models, feature importance was calculated for both PCA and non-PCA cases. The estimates of the importance of the features are shown in Fig. 8. Given that the AIRS-OMI analysis is based on the spectral windows only, AIRS-OMI is not assessed in this section.







**Figure 8.** Feature importance of the CrIS full band models. The left column shows results for the models including PCA, top to bottom, ozone, CO, and TATM. The right column shows the results for the models without PCA, with the same target ordering in rows. The red dots indicate the spectral windows of the retrievals depending on the trace gas, and the black dots indicate wavelengths outside of the spectral windows.

The results shown in Fig. 8 indicate limited differences between the target ML models in the PCA cases, suggesting that the PCAs of each of the ML models focus on similar frequencies. The feature importance for the PCA cases shows that spectral regions far outside of the highlighted spectral windows have a significant impact on the ML model performance. For example, in the case of both Ozone and TATM, features >x2 larger in magnitude are apparent in the SW region of the CrIS spectrum, while neither ozone nor TATM have spectral windows in this region. Note other regions of significant importance outside of defined spectral windows, below 750 cm$^{-1}$ (for Ozone and CO) and between 1500-1750 cm$^{-1}$. Considering the non-PCA



cases, larger deviations between feature importance in the ML models are apparent. For example, the CO case shows significant importance in the SW band, far in excess of the CO spectral window, which is not apparent in the PCA case. However, the TATM case generally shows importance in the same spectral region as the TATM spectral windows, while also exhibiting some importance outside of the fit windows. Note, that there remain similarities between the PCA and non-PCA cases, for example $<750\,\mathrm{cm}^{-1}$. These results suggest that non-fitted elements in the retrieval process have significant impact on the overall quality of retrievals, and potentially hint at some of the underlying reasons behind retrieval failure.

## 4.3 Multiple Flags Performance

The results indicated thus far give clear quantitative evidence that predicting poor-quality retrievals with CrIS and AIRS-OMI is feasible. However, these results are based on training on a single master quality flag which is based on numerous different factors. Some of these factors may have more influence over the master quality flag than others. Meaning, even with the feature importance identified in Fig 8 it is challenging to determine the causes of the failures. Therefore, it is important to identify whether similar results can be obtained by training on the individual quality flags identified in Table 4, and if the influence of these flags can be traced to a specific spectral region. Some of the constituent parts of the master flag may not contribute significantly, and therefore training on the individual flags could result in improved performance. Therefore we performed the same analysis as described previously on each of the relevant flags.



**Table 6** AUC values for each of the quality flags identified in Table 4. Training procedure is the same as identified in Figs. 2 and 3. AUC values are shown with PCA both applied and not applied, for both CrIS and AIRS-OMI cases.

| Flag | Target | Window PCA CrIS \| AIRS-OMI | Window No-PCA CrIS \| AIRS-OMI | Full Band PCA CrIS | Full Band No-PCA CrIS |
|---|---|---|---|---|---|
| Normalised Residual radiance RMSE | O3 | 0.956 \| 0.801 | 0.876 \| 0.842 | 0.918 | 0.909 |
| | CO | 0.623 \| 0.897 | 0.718 \| 0.773 | 0.697 | 0.718 |
| Absolute residual radiance mean | O3 | 0.808 \| 0.803 | 0.745 \| 0.785 | 0.783 | 0.773 |
| | CO | 0.704 \| 0.835 | 0.656 \| 0.799 | 0.794 | 0.782 |
| Absolute value of $\mathbf{K} \cdot \Delta \mathbf{L}$ | O3 | 0.772 \| 0.781 | 0.716 \| 0.831 | 0.760 | 0.753 |
| | CO | 0.715 \| 0.592 | 0.673 \| 0.637 | 0.780 | 0.782 |
| | TATM | 0.773 \| 0.837 | 0.739 \| 0.844 | 0.746 | 0.758 |
| Absolute value of $\mathbf{L} \cdot \Delta \mathbf{L}$ | O3 | 0.709 \| 0.854 | 0.652 \| 0.865 | 0.702 | 0.678 |
| | CO | 0.809 \| 0.674 | 0.750 \| 0.648 | 0.852 | 0.746 |
| Cloud top pressure | O3 | 0.964 \| 0.939 | 0.937 \| 0.950 | 0.966 | 0.972 |
| | CO | 0.916 \| 0.937 | 0.903 \| 0.894 | 0.965 | 0.969 |
| | TATM | 0.968 \| 0.924 | 0.969 \| 0.944 | 0.957 | 0.963 |
| Mean cloud optical depth | O3 | 0.898 \| 0.939 | 0.850 \| 0.950 | 0.868 | 0.875 |
| | CO | 0.788 \| 0.927 | 0.722 \| 0.925 | 0.789 | 0.797 |
| | TATM | 0.841 \| 0.985 | 0.742 \| 0.977 | 0.818 | 0.792 |
| Cloud optical depth variability | O3 | 0.888 \| 0.758 | 0.824 \| 0.821 | 0.763 | 0.752 |
| | CO | 0.885 \| 0.884 | 0.887 \| 0.860 | 0.871 | 0.919 |
| | TATM | 0.855 \| 0.885 | 0.836 \| 0.895 | 0.822 | 0.840 |
| Ozone continuum curve | O3 | 0.941 \| 0.722 | 0.885 \| 0.768 | 0.923 | 0.904 |
| Ozone tropospheric consistency | O3 | 0.844 \| 0.685 | 0.720 \| 0.746 | 0.821 | 0.802 |
| Ozone column error | O3 | 0.962 \| 0.732 | 0.955 \| 0.782 | 0.962 | 0.963 |
| Cloud Fraction | O3 | N/A \| 0.873 | N/A \| 0.926 | N/A | N/A |



The difference between the retrieved surface temperature and the initial a priori surface temperature was not found to be a
useful predictor and, therefore, was not included in this analysis.

For CrIS AUC values, the results shown in Table 6 show a pattern similar to that shown in Table 5, that is, typically the
model trained on the spectral window using PCAs yields the best results. There are some exceptions to this, notably the ML
model trained with the full band without PCA has the best performance in the Cloud top pressure case for ozone and CO. As
with the master quality flag, the training methods (e.g. PCA or non-PCA) have different impacts depending on the quality flag.
For example, the ozone tropospheric consistency flag shows a 15% difference in AUC value, between PCA and non-PCA cases
(when only using the spectral window). While the ozone column error flag shows only a maximum 0.7% difference between
PCA and non-PCA cases. This is also true between targets for the same failure flag, for example, Mean cloud optical depth,
when ozone is the target there is a 5.5% difference between training methods, while for TATM there is a 12.5% difference.
This is less surprising given the differences in the quality range for this flag as identified in Table 4.

In general, training on the individual quality flags yields improved results than training on the master quality, with cloud top
pressure and ozone column error yielding AUC scores of nearly 1 for the window PCA case. Therefore this suggests the ML
models can accurately predict failures in those cases. However, there are some cases where the ML models do not perform as
well as the master quality, for example, the absolute value of $L \cdot \Delta L$. There are multiple reasons for this poorer performance,
for example, $L \cdot \Delta L$ and $K \cdot \Delta L$ may be challenging for the ML models to effectively learn, or the quality ranges defined in
Table 4 could be insufficient, and require further tuning. We note that $K \cdot \Delta L$ has the largest failure rate in Table 4, which
logically would mean the ML model should have more information about this flags failures as opposed to others, yet the AUC
scores suggest otherwise. However, we note that other flags (e.g. ozone column error), also have high failure rates, but better
ML model performance, meaning failure rates are unlikely to be influencing the ML model performance. In the case of CO, the
ML models are challenged with Normalised Residual radiance RMSE and the Absolute residual radiance mean, most likely
due to the extremely short CO spectral window (Table B3).

Figure 8 showed the importance of different spectral regions on the master quality flags for CrIS retrievals, and Fig. 9 shows
the same analysis for each of the individual quality flags. In this case, we are not investigating the PCA-based ML model
results, since these show the same patterns as Fig. 8, i.e., importance at the same spectral locations, independent of the flag.
It is important to identify that for most cases, the best ML model results are achieved by a combination of PCA in the ML
pipeline, and training on the spectral window only (at least in the case of ozone). However, in the following analysis, we aim to
identify any potential influence of spectral regions outside of the immediate spectral windows. Such information may influence
future spectral window choices and indicate how the whole measurement may affect retrieval failures.




**Figure 9.** Feature importance for each of the individual quality flags, based on an ML model not including PCA, trained on the full CrIS band. The left-hand column shows results from ozone, the middle column CO and the right-hand column TATM. Each row refers to a different flag, as identified in the panel title. Gaps in the panels indicate where a flag is not used for the relevant target. The spectral window of the target is highlighted in red on each panel.





The residual radiance RMS flag feature importance results are shown in row 1 of Fig. 9, there is clear dependence on the CrIS SW for all three targets, with the spectral windows for all three targets appearing to be relatively unimportant. Similar to the master quality flag, failures due to RMS residual radiance could be caused by a number of reasons, e.g. clouds in the light path or poor estimation of scattering. Meaning that attributing RMS failures to specific causes will be difficult. The CrIS retrieval pipeline includes retrieval of cloud top pressure and extinction, which include spectral windows in the SW, meaning it may be possible to attribute this sensitivity to cloud-related failures. Note the feature importance plots for cloud top pressure (row 5) and average cloud optical depth show similar behaviours in the SW. Focusing on mean residual radiance in row 2, for ozone and TATM importance features are evident in the LW, which match some of the spectral windows of TATM, implying mean residual radiance for ozone is dependent on TATM. Conversely in the SW, CO and TATM show similar features, suggesting a CO dependence on TATM in the SW. The feature importance for $K \cdot \Delta L$ (row 3) in the case of CO and TATM is very similar to the equivalent plots for mean residual radiance. Indeed the AUC values in Table 6 are very similar for CO. Implying that these quality flags draw information from the same spectral regions. However, there is less similarity in ozone, where far more importance is attributed to the micro windows in the LW and MW. However, significant importance is still apparent in the SW, again suggesting ozone absorption outside of the CrIS MUSES micro windows can impact the quality criteria. The importance of features with the $L \cdot \Delta L$ flag is shown in row 4, with only ozone and CO using this flag. The ozone micro windows do not show significant features and are similar to the results shown for the mean residual radiance, implying that the whole CrIS spectral range contributes to this failure flag. This is contrasted by the feature importance for CO, where the LW and MW have lower levels of importance when compared to the SW.

The cloud top pressure flag in row 5 shows similar features for all three targets, with notable features at 750 cm$^{-1}$, 1000 cm$^{-1}$, 1500 cm$^{-1}$, 2100 cm$^{-1}$ and 2400 cm$^{-1}$. Cloud top pressure is one of the few flags that shows the best performance when trained on the whole spectral range, which is highlighted by the fact that the feature importance plots are almost identical across the targets. Note that the allowable range for Cloud top pressure is identical for all three targets. Row 6 shows the features for average cloud optical depth, again there are similarities between the CO and TATM features, with the maximum importance towards the shorter end of the SW. Ozone, while having similar features to CO and TATM in the LW and MW, shows unique characteristics in the SW. Here, both CO and TATM have identical quality criteria for average cloud optical depth, while ozone has a much more stringent requirement. Note the ozone windows in the LW indicate significant importance, which is supported by the AUC values in Table 6, where there is limited difference in the AUC values, despite the learning method. Implying cloud optical depth is best described by the ozone windows in the LW. Row 7 shows the feature importance of cloud variability, in this example, each of the targets exhibits very different behaviour. For ozone, the feature importance is largely equivalent across the whole spectrum, suggesting that no information is gained outside of the ozone spectral window, which is supported by the AUC values. For CO, there is significant importance shown in the short end of the SW, similar to cloud top pressure. The AUC values in Table 6 for CO suggest that improved ML model performance by using the whole available spectral band, suggesting that additional CO windows not used in the MUSES retrievals are possible. For TATM there is little variation between the bands, and the AUC values do not indicate significant differences between the learning methods, thus implying there is nothing to learn from the wider spectral bands. The final three flags in rows 8, 9, and 10 are only relevant to





ozone. For ozone continuum curvature, the feature importance is similar to that of CO with cloud variability. In general, there is low feature importance, however, the SW which has no ozone windows indicates two spectral regions where importance is

larger than any other feature. The MW channel generally shows no importance, except where the ozone micro-windows are found. While the LW shows importance across the whole band. For the ozone tropospheric consistency flag in row 9, limited importance is attached to the ozone micro windows, with the largest features occurring towards the shorter end of the MW and the longer end of the LW. Finally, the ozone column error is investigated on row 10, note from Table 6 that the AUC values for each learning method are very similar, suggesting that the importance is largely confined to the ozone micro windows. The

feature importance plot largely supports this result, with the majority of the features confined to the ozone micro windows, or the surrounding spectral regions.

As with the feature importance plots for the master quality flags shown in Fig. 8, it is not possible to identify one spectral region as the cause of flag failures. Although in some cases it is more obvious than others, for example, flags such as RMS residual radiance are dependent on numerous effects, while ozone column error is restrained to certain spectral regions. There

is some indication in these results that this type of feature analysis could be used to further refine spectral windows for trace gas retrievals. Further, the SW CrIS band, despite having limited use in the MUSES CrIS retrievals (CO, and a cloud micro window), seems to have significant importance across most of the failure flags. Further investigation into why this is the case is required.

## 5 Statistical analysis using independent dataset

The previous subsections have quantified the performance of the ML models, however, in practice, a threshold value must be chosen in order to apply the ML models. Here we analyse the statistical significance of the ML model predictions by relating the binary predictions to the true failure flags output from the MUSES algorithm. Using an independent CrIS dataset not used to train the ML models, in this case roughly 40000 retrievals from August 12th, 2020. Figure 11 compares the MUSES failure flags with the percentage probability of failure predicted by the ML model trained using PCA on the spectral windows alone.



**Figure 10.** Quality flags from MUSES CrIS retrievals of CO, TATM, and O$_3$ on January 1st 2020, where red indicates triggering of the quality flag (i.e. "bad quality") and green "good quality"; left-hand column. Predicted probability of failure [0-1] from ML models for CO, TATM, and O$_3$ on the same day; right-hand column

What is clear from the analysis of CrIS data in Fig. 11 is that the ML models typically do a good job of predicting the actual failures. However, in the locations surrounding the failure positions, the ML models often predict a high probability of failure, reducing the performance of the model, meaning that the choice of the threshold will have a significant impact on the use of the ML models.





**Figure 11.** Quality flags from MUSES AIRS-OMI retrievals of CO, TATM, and $O_3$ from a day in 2020, where red indicates triggering of the quality flag (i.e. "bad quality") and green "good quality"; left-hand column. Predicted probability of failure [0-1] from ML models for CO, TATM, and $O_3$ on the same day; right-hand column

To assess the importance of this threshold, we use Cramér's V metric to assess how strongly two categorical variables (in
this case the reported MUSES failure flag and the ML model predicted failures) are associated. With this analysis, we can understand if there is any statistical significance between what the ML models predict as failures and the truth. Cramér's V metric is defined as:



$$V = \frac{\sqrt{\frac{\chi^2}{n}}}{DOF},\tag{2}$$

where $\chi$ is the Chi-square statistic, n is the total sample size and DOF is the degrees of freedom of signal of the dataset. A

value of 0 for V means that there is no association and 1 means perfect association; however, the interpretation of the degree of association depends on the DOF, which in this study are equal to 1. In this case, we assume a small association is $0.1 \leq V < 0.3$, a medium association is $0.3 \leq V < 0.5$ and a large association is $V \geq 0.5$.

Figure 12 indicates the Cramér's V metric for the CrIS dataset for each quality flag for the three target quantities for a range of ML model thresholds. Both ozone and CO show peak importance at the threshold value of 0.6, while also showing a steady

increase in importance between thresholds of 0.1 to 0.6. The reasons why ozone and CO have a maximum association at 0.6, but TATM is at 0.4 are unclear, but most likely due to the different quality criteria. Note that each quality flag for CO shows similar importance values at each threshold, while TATM and ozone show more variation. For example, the master quality flag and ozone column error show the strongest associations, even at high thresholds, unlike any of the remaining quality flags. This is an interesting result, given that Table 6 shows numerous ozone quality flags as having AUC values similar to ozone column

error.





**Figure 12.** Cramér's V statistics for the CrIS dataset between quality flags and the independent predicted dataset for varying thresholds of pass/failure ranging from 0.1-0.9 for the three gases; CO, TATM, and O3. A small association is defined by values 0.1-0.3, a medium associated 0.3-0.5, and a large associated >0.5.

Figure 13 displays the Cramér's V metric for the AIRS-OMI dataset for each quality flag for the three target quantities for a range of ML model thresholds. Both TATM and CO show slightly higher importance for the quality flag above other flags, with medium associations at 0.2-0.3. For CO, peak importance is achieved at a threshold of 0.8 and for TATM remains relatively constant with threshold, but a peak at 0.9. All other flags for CO reach peak importance at a threshold of 0.5. For TATM, associations are small, ranging from 0.1-0.3, and show no clear pattern between flags or threshold. The only exception is CloudVariability_QA, which has a higher importance at low thresholds. Ozone flags show the strongest associations compared



to TATM and CO. In particular, the Quality_Flag, RadianceResidualRMS and OMI_CloudFraction flags display strongest associations above 0.5. In the case of the latter flag, importance is constant with threshold (always greater than 0.6) indicating that Cloud Fraction is a critical quality flag despite prediction threshold. All other flags show decreasing importance with
increasing thresholds, starting with relatively high associations at 0.5.

**Figure 13.** Cramer's V statistics for the AIRS-OMI dataset between quality flags and the independent predicted dataset for varying thresholds of pass/failure ranging from 0.1-0.9 for the three gases; CO, TATM, and O3. A small association is defined by values 0.1-0.3, a medium associated 0.3-0.5, and a large associated >0.5.

Figure 14 shows the result of applying the ML filtering technique using threshold values of 0.5 and 0.2 to the MUSES CrIS retrieval pipeline for ozone. The CrIS retrieved ozone concentrations at an exemplar pressure level (681 hPa), split into daytime



and nighttime. The top panels show the retrievals without quality control or ML filtering to act as a baseline, where a total of 39892 retrievals are available. The middle panels show the CrIS retrievals with MUSES quality control applied, where a pass rate of 73% is found. The second from the bottom panels indicates MUSES retrievals with ML filtering of threshold 0.5 and quality control applied, where after filtering 28085 retrievals are available, with a pass rate of 85%. The bottom panels indicate a filtering threshold of 0.2 (i.e. retrievals with a probability of failure of 0.8 or above are removed). In this case, 37683 retrievals are available, with a pass rate of 77%. Once adjusted for quality, the non-ML case has 29121 good quality retrievals, the ML threshold of 0.5 case has 23872, and the threshold of 0.2 case has 29015. These results show a clear indication of the impact of the ML model. For the 0.5 threshold case, the ML model removes 35% of the retrievals in the standard pipeline, meaning in this case a speed up of a third is achieved. In the 0.2 threshold case 6% of the retrievals are removed, whilst retaining a similar number of good quality retrievals to the non-ML case.

In the threshold 0.5 case, the speed-up of 35% is a huge gain, however, there is a cost, a 20% loss in good quality retrievals. This loss is obvious in Fig. 14, with clear patterns to the filtered retrievals. The majority are removed from the Sahara desert, the Arabian peninsula, central Asia, and the western United States. The retrievals that have been removed are typically on the extreme end of magnitude, especially over central Asia. We note that when quality control is applied to the CrIS retrievals, it is largely the high-concentration values that a removed. Implying high ozone concentration retrievals in the troposphere are more likely to be poor quality, and is likely the reason why the ML model classifies high ozone concentrations as more likely to fail. The mean ozone concentration for the non-quality controlled case is 47 ppb, the quality controlled case is 46 ppb and for the threshold 0.6 ML case 45 ppb. For the threshold of 0.2 case, the speed up is not as dramatic, saving roughly 3 hours of processing (assuming multiple cores and multiple threads) in the shown case. However, over larger time periods, such time saving will mount up quickly.

This challenge of failures over desert regions requires additional analysis, it is possible that more effective results will be obtained by training an ML model with only data taken over deserts. Suggesting that regional ML models may be more effective than global models.



**Figure 14.** Impact of applying the ML filter to MUSES ozone retrievals. The top panels show CrIS daytime and nighttime retrievals at the 681 hPa pressure level using the standard MUSES processing with no quality control. The middle panels are as above, but standard quality control flags are applied. The bottom panels show the same data when the ML filter is applied with a threshold of 0.5 and 0.2, and the remaining data has been quality controlled.





Figure 15 indicates the results of applying ML filtering technique using threshold values of 0.5 and 0.2 to the MUSES AIRS-OMI retrieval pipeline, as with the CrIS case for ozone. The top left panel show the retrievals without quality control or ML filtering to act as a baseline, where a total of 24382 retrievals are available. The top right panel shows the AIRS-OMI retrievals with MUSES quality control applied, where 8026 good quality retrievals are available, indicating a pass rate of 33%.

The bottom left panel indicates MUSES retrievals with ML filtering of threshold 0.5 and quality control applied, where 6386 good quality retrievals are available, meaning the ML filtering captures 80% of the good quality retrievals. The bottom right panel indicate a filtering threshold of 0.2. In this case, 7850 good quality retrievals are available, meaning the ML filtering capture 98% of the good quality retrievals. These results show a clear indication of the impact of the ML model. For the 0.5 threshold case, the ML model removes 74% of the retrievals in the standard pipeline, meaning in this case a speed up of a 3/4

is achieved. In the 0.2 threshold case 68% of the retrievals are removed. Unlike in the CrIS case, for AIRS-OMI Fig 15 shows the ML filtering does not target and remove specific geographical regions, indicating the cloud filtering works very well, while offering huge speed ups in processing.

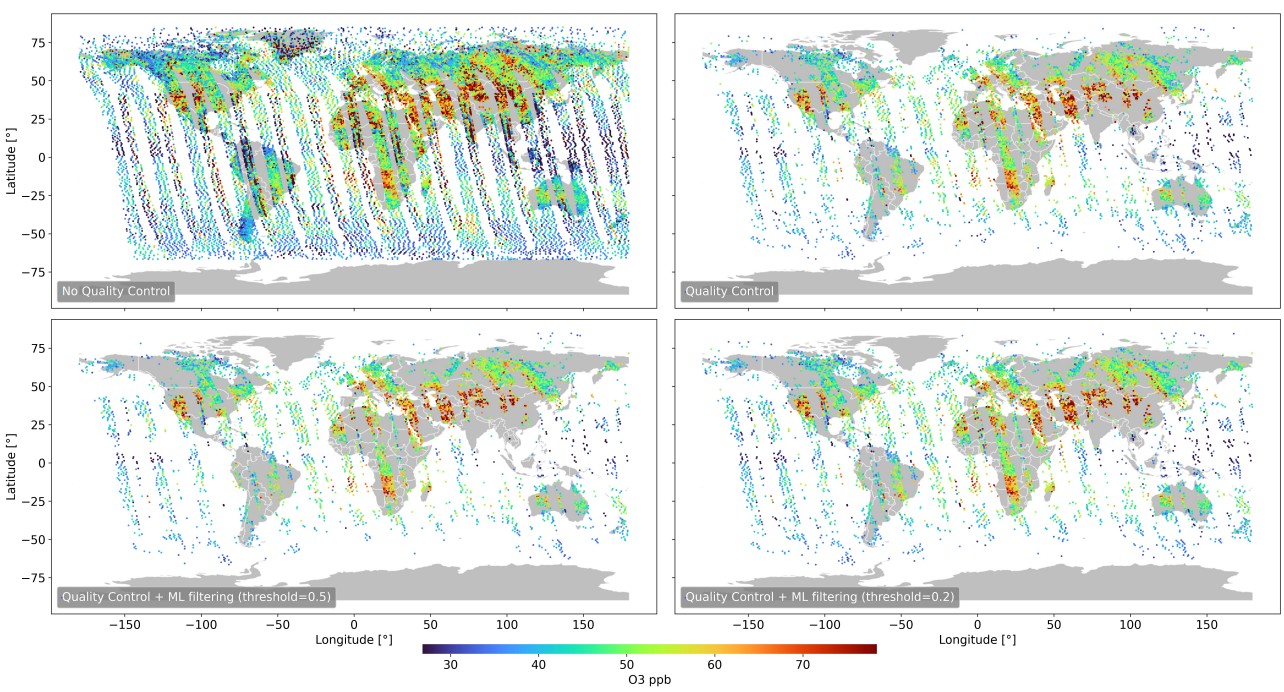

**Figure 15.** Impact of applying the ML filter to MUSES ozone retrievals for AIRS-OMI. Daytime retrievals at 681 hPa pressure level using the standard MUSES processing with no quality control (top left) and standard quality control flags applied (top right). The ML filter is applied with a threshold of 0.5 (bottom left) and 0.2 (bottom right) and remaining data has been quality controlled.



## 6   Discussion

One of the primary metrics used in this paper AUC, is an efficient way of assessing model performance. However, a choice
must be made when implementing a failure prediction model in a retrieval pipeline. For example, should careful use of the
models be employed, i.e., setting the failure threshold value high and only removing retrievals that have a very high probability
of failure, but still allowing a significant percentage of retrievals that will fail through the pipeline. Or should no caution be
used, setting the failure threshold low and removing almost all of the failed retrievals, but also removing large volumes of good
quality retrievals as well. There are arguments to be made for both positions, however, currently, it is not practically possible
to process the millions of satellite measurements and convert them into L2 trace gas concentrations in real-time. Therefore, if
as much real-time data as possible is desired, the most logical solution will be to use a low threshold, therefore removing most
of the available data from the retrieval pipeline, but guaranteeing a high likelihood that all processed retrievals will be of good
quality.

The threshold values lead to a further point of contention, the MUSES CrIS and AIRS-OMI retrieval pipelines simultane-
ously retrieves CO and ozone, as well as several other trace gases. There will be cases where the ozone retrieval will fail, but
other products may not, or vice versa. In which case a decision must be made whether or not to ignore all trace gas retrievals
from a particular spectrum, or keep those that do pass the initial failure check.

There is a significant cost/benefit aspect to the ML model at this time. Where significant processing speed-ups can be
achieved, but potentially valuable information may be lost. This cost/benefit can be improved with more training of the ML
model, potentially to the point where there is very little cost in applying the ML model. This work represents the first step
in understanding why and how retrievals fail, future work can focus on learning the particular combinations of conditions
that cause retrievals to fail, and modify the algorithms to take into account these factors and thus reducing the likelihood of
producing failed retrievals.

In general, training is key to the effectiveness of ML models. Numerous training datasets were applied, including much
denser sampling of CrIS retrievals, yielding dataset sizes of 100,000 retrievals plus. However, in general, we found minimal
impact, both in AUC scores and experiments similar to Fig. 14. This highlights a challenge, given the difficulty experienced by
the ozone ML model over desert regions, indicating that blindly training with larger datasets will not solve the problem. Some
ways to address this would be by taking into account the fail-rate of different regions when preparing the training dataset, or
by taking into account geographical location when performing over/under-sampling.

As satellite instruments age, the quality of the spectral radiances can degrade. For example in the case of OMI, the quality
of some OMI pixels has limited the latitudinal range of the instrument (Levelt et al., 2018). While in the case of Suomi NPP
CrIS, failures in the long-wave channels of the 'side 2' electronics suite in May 2021 forced a switch to 'side 1' electronics in
order to retain the use of the LW channels (Iturbide-Sanchez et al., 2021). The implication is that as instruments age and decay,
the ML models will need to be re-trained to account for any degradation.



## 7  Conclusions

The ability of retrieval algorithms to convert satellite spectra into trace gas quantities in a timely manner is a key challenge in the future of EO. Tens of millions of measurements will be generated per day, representing a significant challenge to process all of these measurements in real-time. A significant drain on the processing of these millions of retrievals is that failed retrievals require equal resources to good quality retrievals, wasting huge amounts of computational effort. In this paper, we provide an ML method for reducing the processing overhead of retrieval algorithms, by predicting whether or not a retrieval will fail based on the characteristics of an instrument measured spectrum, prior to performing a full retrieval. This was achieved by training an extraTrees ML model on Suomi NPP CrIS and AIRS-OMI spectra and quality flags from the TROPESS/MUSES algorithm for ozone, carbon monoxide, and temperature profile retrievals. We show a test case focusing on ozone, where from a pipeline of 37683 CrIS retrieval targets, applying the ML filter prior to full retrieval removes 13811 targets, yielding a speed up of 35%. Of the 13811 targets, $\sim$20% were miss-classified, which can be reduced given more targeted training regimes. While in the case of AIRS-OMI from a pipeline of 24382 retrieval targets, the ML filter removes 16532 targets, yielding a speed up of 66%. Of the 16532 targets, $\sim$2% were mis-classified, showing a high quality tool.

The retrieval algorithm quality flags used in this assessment are based on numerous individual flags, designed to catch errors. We show that in some cases, specific spectral regions can be identified as influencing these flag failures. Focusing on these spectral regions could help identify why retrieval failures occur.

The ML models identified in this paper are based on open-source python packages which are simple to train and apply, given sufficient training data. This failure prediction model represents a significant contribution towards reducing the processing overheads of current and future EO satellites.

*Code and data availability.*  The code used to train the ML models are available at https://github.com/brencej/RetrievalFailure. The training datasets used for this study are available upon discussion with the lead author. Use of the MUSES algorithm is based on discussion with the lead author. TROPESS/MUSES data products are available at https://tes.jpl.nasa.gov/tropess/.





## Appendix A: TROPESS CrIS and AIRS-OMI State Vectors

**Table A1** List of parameters in the MUSES/CrIS retrieval state vector for ozone, carbon monoxide, and temperature profile. Sources of the a priori and covariance are included.

| Instrument | Fitting parameter | Target | Number of parameters | A priori |
|---|---|---|---|---|
| AIRS-OMI,OMI,AIRS,CrIS | $O_3$ at each pressure level | $O_3$, TATM | 25 | MOZART-4 |
| AIRS,CrIS | $H_2O$ at each pressure level | $O_3$, TATM | 16 | GEOS-5 |
| $N_2O$ at each pressure level | TATM | 25 | | |
| AIRS,CrIS | $CH_4$ at each pressure level | TATM | 25 | MOZART-3 |
| AIRS,CrIS | CO at each pressure level | CO | 14 | MOZART-3 |
| AIRS,CrIS | Temperature profile | TATM | 32 | NCEP |
| AIRS,CrIS | Surface temperature | $O_3$, CO, TATM | 1 | GEOS-5 |
| AIRS,CrIS | Surface emissivity | $O_3$, CO, TATM | 19 | UOW-M database |
| AIRS,CrIS | Cloud extinction | $O_3$, CO, TATM | 10 | Initial Brightness Temperature Difference |
| AIRS,CrIS | Cloud top pressure | $O_3$, TATM | 1 | 500 mbar |
| AIRS-OMI,OMI | Cloud Fraction | $O_3$ | 1 | Derived from 346.5-347.5 nm |
| AIRS-OMI,OMI | Ring Scaling | $O_3$ | 1 | 1.9 |
| AIRS-OMI,OMI | Irradiance shift | $O_3$ | 1 | 0.0 |
| AIRS-OMI,OMI | Radiance/Irradiance shift | $O_3$ | 1 | 0.0 |
| AIRS-OMI,OMI | Albedo (zero order) | $O_3$ | 1 | OMI climatology |
| AIRS-OMI,OMI | Albedo (first order) | $O_3$ | 1 | 0 |
| AIRS-OMI,OMI | Albedo (second order) | $O_3$ | 1 | 0 |



The sources for the a priori profiles are Model for OZone and Related chemical Tracers (MOZART)-3 and 4 (Brasseur et al., 1998; Park et al., 2004; Emmons et al., 2010) for the ozone, methane, and $N_2O$ profiles and covariance. Water vapour, temperature profile, and surface temperature data from the Goddard Earth Observing System Model, Version 5 (GEOS-5) (Suarez et al., 2008) and covariance from the National Center for Environmental Prediction (NCEP) reanalysis (Kalnay et al., 1996). The emissivity a priori is taken from the University of Wisconsin-Madison (UOW-M) Global infrared land surface emissivity database (Seemann et al., 2008). The spectroscopic parameters for the target gases and interfering gases come from the High Resolution TRANSmission (HITRAN) 2012 database (Rothman et al., 2013).

## Appendix B: Retrieval windows

The spectral windows for the targets covered in this study for AIRS and CrIS are highlighted in Table B1 for ozone, Table B3 for carbon monoxide, and Table B4 for temperature profile. For the OMI ozone window, Table B2. These are graphically represented in Fig. 1.

**Table B1** MUSES micro windows used for CrIS/AIRS ozone retrievals

| CrIS/AIRS Band | Window Start [cm$^{-1}$] | Window Stop [cm$^{-1}$] | Species |
|---|---|---|---|
| Band 1 (LW) | 950.00 | 1031.25 | H2O, HDO, NH3, O3, CO2 |
| Band 1 (LW) | 1043.125 | 1048.75 | H2O, HDO, NH3, O3, CO2 |
| Band 1 (LW) | 1068.75 | 1088.75 | H2O, HDO, NH3, O3, CO2, CH4, CFC11, CFC12 |
| Band 1 (LW) | 1094.375 | 1095.00 | H2O, CH3OH, HDO, NH3, O3, CO2, CH4, CFC11, CFC12 |
| Band 2 (MW) | 1211.25 | 1215.00 | H2O, HDO, O3, CO2, CH4, N2O |
| Band 2 (MW) | 1223.75 | 1227.50 | H2O, HDO, O3, CO2, CH4, N2O |
| Band 2 (MW) | 1258.75 | 1261.25 | H2O, HDO, O3, CO2, CH4, N2O |
| Band 2 (MW) | 1265.00 | 1267.50 | H2O, HDO, O3, CO2, CH4, N2O |
| Band 2 (MW) | 1268.75 | 1271.25 | H2O, HDO, O3, CO2, CH4, N2O |
| Band 2 (MW) | 1311.25 | 1317.50 | H2O, HDO, O3, CO2, CH4, N2O |



**Table B2** MUSES micro windows used for OMI ozone retrievals

| CrIS Band | Window Start [cm$^{-1}$] | Window Stop [nm] | Species |
|---|---|---|---|
| UV1 | 270.00 | 305.00 | O3 |
| UV2 | 312.00 | 329.50 | O3 |

**Table B3** MUSES micro windows used for CrIS/AIRS carbon monoxide retrievals

| CrIS Band | Window Start [cm$^{-1}$] | Window Stop [cm$^{-1}$] | Species |
|---|---|---|---|
| Band 3 (SW) | 2181.25 | 2200.00 | H2O, O3, CO2, N2O, CO |

**Table B4** MUSES micro windows used for CrIS/AIRS Temperature profile retrievals

| CrIS Band | Window Start [cm$^{-1}$] | Window Stop [cm$^{-1}$] | Species |
|---|---|---|---|
| Band 1 (LW) | 671.25 | 728.75 | H2O, CO2, O3, N2O, HNO3, CFC11, CFC12, CCL4, CFC22, NH3 |
| Band 1 (LW) | 732.50 | 780.00 | H2O, CO2, O3, HNO3, CFC11, CFC12, CCL4, CFC22, NH3 |
| Band 1 (LW) | 810.00 | 901.875 | H2O, CO2, O3, HNO3, CFC11, CFC12, CCL4, CFC22, NH3 |
| Band 2 (MW) | 1210.00 | 1250.00 | H2O, CO2, O3, N2O, CH4, HDO, HNO3, CFC12 |
| Band 2 (MW) | 1252.50 | 1264.375 | H2O, CO2, O3, N2O, CH4, HDO, HNO3, CFC12 |
| Band 2 (MW) | 1266.25 | 1300.0 | H2O, CO2, O3, N2O, CH4, HDO, HNO3, CFC12 |
| Band 2 (MW) | 1307.50 | 1317.50 | H2O, CO2, O3, N2O, CH4, HDO, HNO3, CFC12 |

*Author contributions.* EM conceived of the study, generated MUSES retrievals, and wrote the paper. JB and JT designed and evaluated the
510    ML model. JA performed the Cramér's V analysis on the independent dataset and generated the figures. VK developed the major components
of MUSES, EM, SD and KWB guided and managed the research. All authors reviewed the paper.



*Competing interests.* The first author is a moderator for EGUsphere, all other authors declare no competing interests.

*Acknowledgements.* Thanks to Vijay Natraj at JPL for reviewing the paper. A portion of this work was carried out at the Jet Propulsion Laboratory, California Institute of Technology, under a contract with the National Aeronautics and Space Administration (80NM0018D0004).



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
