# Peer review of "Predictions of satellite retrieval failures of air quality using machine learning"

_EGUsphere, 2024_

## Author Comment (AC1)

Response to Reviewer 1

Thank you for the review, we respond to your comments below. We have kept your original text in black, our responses are in blue and specific changes are in underlined blue.

In their manuscript "Predictions of satellite retrieval failures of air quality using machine learning", the authors report on a study aiming at reducing the computational load of satellite data retrievals by identifying measurements for which the retrieval has a high probability of failing before even starting the retrieval. Their proposed method is based on a machine learning approach, trained on a set of satellite spectra and the corresponding error flags from the retrieval. The technique is demonstrated on MUSES retrievals of CO, temperature profiles and ozone using CrIS and AIRS/OMI radiances. The results show that a large fraction of unsuccessful retrievals can be avoided by applying the ML filtering, and that only a moderate number of successful retrievals is skipped.

The topic of this study is relevant as slow optimal estimation type algorithms can often not be applied to all measurements of modern satellite instruments due to a lack of computational resources. Reducing the number of unsuccessful retrievals, therefore helps in providing a larger number of results at the same cost. The proposed method is convincing, and the manuscript is clearly written and structured. However, I have several concerns and suggestions which should be considered before the manuscript can be accepted for publication.

**Major comments**

1) When using machine learning approaches in science and data retrieval, there is always the same concerns:

- Are all relevant situations covered appropriately in the training data set?

While we have tried to cover as much as possible the range of conditions the satellites will encounter, by covering as much as possible seasonal and spatial ranges. The reality is there are likely to be situations that won't have been covered by the training set. The results from this study show already good performance, which could be improved with constant updated training of the ML model to take into account situations not originally covered.

- Has the method generalised the information sufficiently to be applied to another data set?

That depends on the new data set. Section 5 presents an evaluation of the trained models on a completely new, so-far unused dataset from CrIS. The results of this

evaluation show no significant drop in prediction performance on a new data set. This indicates that the model generalizes well to unseen data.

On the other hand, we have not tested the generalization on new data sets generated using different retrieval algorithms or featuring significantly different spatial or temporal distributions. It is reasonable to expect a drop in prediction performance when the test set has properties significantly different from the training set. This is alleviated by the simplicity and quick training time of the applied method, which makes it easy to adapt and train on such data sets.

- Did the algorithm learn the intended connections, or has it generalised correlations which exist by chance or are not cause-and-effect type links?

This is a challenging question. While some causes for poor quality retrievals are understandable, the application of machine learning allows models to exploit complex patters in the captured spectra that are connected to retrieval failure. Such complex patterns can often be difficult to interpret and analyse. Furthermore, the type of machine learning and model analysis we used in this work gives information only on correlations and connections. Analysing cause and effect is a much more difficult task that typically requires controlled experiments and more sophisticated approaches.

Some insights are given by our analysis of the feature importance of different wavelengths. However, more research is required for actionable insights on the failures of retrieval algorithms.

Nevertheless, we are confident that the models' predictions are not primarily based on spurious correlations. We back this claim with our evaluation procedure. We evaluated the performance in two stages – first, using cross validation (a standard and rigorous evaluation procedure in machine learning) and later using a completely new, so far unused data set. The relatively high predictive of our models indicates that they are capturing meaningful information.

- Is a bias of some form introduced in the results when applying the method?

Yes, some bias is inevitable introduced through the construction of the data set. This is true for all models trained using machine learning. To alleviate this, the trained models should not be applied in conditions significantly different from the training set.

Our analysis in Section 5 also reveals that the frequency of retrieval failure varies significantly geographically. This imbalance unfortunately leads to a bias in the trained models, which may produce more false positives in those regions. To alleviate this in future work, we propose the development of regional, specialized models. Alternatively, stronger global models could be obtained by constructing a geographically balanced training set, ensuring that the frequency of retrieval failure does not vary significantly in different geographical regions.

Some of these questions are discussed throughout the manuscript, for example, in the context of the erroneous flagging of high CO values. However, the manuscript would benefit from a specific section discussing the potential problems of the method and what the authors found in their tests.

We have included some of the text from the previous answers into the discussion to address the reviewers' request. As follows:

"Finally, as with all ML approaches there are challenges that could cause some problems with the results. For example, are the training datasets representative, or are biases introduced during training, or many other common issues not directly identified here. It is likely some issues are present in the current for of the ML model presented in this paper (for example biases). However, in order to increase confidence in the results, we evaluated the performance of our ML model in two stages – first, using cross validation (a standard and rigorous evaluation procedure in machine learning) and later using a completely new, so far unused data set. The relatively high predictive ability of our models indicates that they are capturing meaningful information and are effective. Therefore, although the performance of the ML model most likely can be improved, we are confident that they are effective."

2) As far as reported in the manuscript, the method was tested on a very limited data set. Surely, more MUSES retrievals are available to test fast ML filtering. Can a more robust test be performed using data from different seasons and different years?

This is an important point, our judgement with this paper was that it is already quite long through the description of the method involved, and further extending the paper through additional comparisons would make the paper exceptionally long. The current level of comparisons, which shows comparisons for two different satellites add weight to the validity of the methods.  We therefore propose a follow on paper which future refines the ML tool and provides a much wider comparison would be more appropriate, than extending the current comparisons.

3) The discussion of the results for the individual flags is interesting but confusing to me. I do not understand why a new metric (the Cramer's V metric) is introduced instead of simply using the number of successful predictions and the number of false positives as quality criteria as in other parts of the manuscript. Maybe I just did not understand what the authors tried to achieve, but I do not see the benefit of this discussion.

The Cramer's V statistic allows us to understand the overall strength of association between the successful predictions under different thresholds and the various quality flags at each measurement location. We chose this statistic since it allows us to evaluate association between two categorical or nominal values, and it takes into

account each measurement location having a prediction and various quality flags, then calculates the strength of association over all these locations.

4) In several places, the authors try to use the results of the ML filtering to identify spectral regions linked to certain error flags. This makes sense for cloud-related flags, as different parts of the spectrum contain different kinds of cloud information, and the ML algorithm may identify them. However, the formulations used in the text are sometimes unfortunate; for example, "Further, the SW CrIS band ... seems to have significant importance across most of the failure flags" suggests that a certain spectral region outside the fitting window is the source of a given retrieval problem, while in reality, one condition (such as broken clouds) can lead to effects in different regions. The ML filter does not necessarily hint at cause-and-effect relations but at correlations.

We definitely agree that cause-and-effect relations cannot be directly inferred by analysing the model. In the reviewer's example, it is indeed surprising that a spectral region outside the fitting window should be associated with retrieval failure. However, it is important to note that our analysis does not claim that region is the source of retrieval failure. Instead, it indicates that region is a useful feature for predicting the success of retrieval. Although the retrieval algorithm does not use that region as input, the measurement conditions that affect the retrieval can be visible in the region and used by the model for prediction, as the reviewer correctly surmised.

In other words, our analysis shows spectral regions that are important for model predictions. The best way to interpret this is to consider spectral regions of high importance as promising candidates for further investigations into retrieval failures that could potentially lead to identification of causes.

5) The OMI-AIRS ozone retrieval appears to be a very good example of the large benefits of ML-based data prefiltering. However, simple filtering using the OMI cloud product would be nearly as efficient in a real-world application without any additional machine learning effort. In general, filtering for known problematic or not interesting conditions could probably be a more transparent alternative to the ML filtering approach proposed here.

This is a highly valid comment, for which there is no clear answer at this point. It may turn out that the ML cloud based filter is more effective than using the cloud product, or vice versa, and we think this is an excellent topic for further investigation where a study directly compares the impacts of these tools. We include an additional element in the discussion to cover this point.

**Detailed comments**

- L7: duplication of "measurements"

Corrected, thank you. Changed to "multiple satellites"

- L12: applied to many EO satellites – applied to data from many EO satellites

Changed, thank you.

- Introduction: I do not see why the data rate of GEO instruments should be higher than that of LEO instruments. In practice, this might be the case, but this is more linked to GEO instruments being rather recent additions with better detectors.

This is true, we have removed text that have implied GEO instruments have a higher data rate than LEO instruments.

- Introduction: I think the main message of the authors is that more data is coming from the new generation of satellite instruments than can be analysed in NRT. For this simple statement, many references are used, which does not make sense to me. I suggest reducing them and focusing on those relevant to this study.

We have reduced the number of references as suggested, but we feel it's important to keep a significant number here, as this highlights the wide range of methods and techniques currently under investigation to help with the ongoing problems of speeding up retrievals.

- Introduction: I also think that it should be mentioned that the problem addressed is mainly limited to Optimal Estimation type retrievals, while many other algorithms are fast enough to process the full volume of satellite data routinely.

We have highlighted that this paper largely targets optimal estimation retrieval algorithms.

- L67: "retrievals absorb" => "retrievals use absorption"

Corrected.

- Tables 1 & 2: I do not see the need for these tables

Thank you for this comment, we respectfully disagree as we think these tables provide relevant information about the instruments used in this paper and provide valuable context.

- Section 2.3.1: I think this can be shortened as it is not relevant to the manuscript

A similar comment was made by one of the other reviewers, and accordingly we have made this section shorter.

- Section 3.1.2: I'm not an ML expert, but I think it would be good to add a bit more information on the method of the "Extremely randomised trees" used here – are there no hyperparameters and other settings specific to the model you applied?

We added the following text to expand on the model details:

"In our experiments, we used the Scikit-learn implementation of extremely randomised trees with 100 trees in the ensemble, no depth limitation, Gini impurity as the measure of split quality, requiring at least 2 samples to split a node and at least 1 sample in leaf nodes. The rest of the hyperparameters were left at their default values."

- L247: "only training is performed on" => "training is performed only on"

Corrected, thank you.

- L279: "These results suggest that non-fitted elements in the retrieval process have a significant impact on the overall quality of retrievals" – I'm not sure what the authors are trying to say and how this can be deduced from the fact that the ML algorithm is using information from outside of the fitting window to predict failure of the retrieval better. To me, this feels like a confusion of correlation and cause-and-effect relationship.

We have softened this statement to read as follows:

"These results suggest further investigation into non-fitted elements in the retrieval process, as these may be having an impact on the overall quality of retrievals, and potentially hint at some of the underlying reasons behind retrieval failure."

- L308: As mentioned above, the ozone failure flag is special as it is linked to cloud cover in a simple and easy-to-predict way.

We are not sure how to respond to this comment.

- Section 4.3: I was surprised that the authors did not evaluate whether combining the prediction of individual flags would be better than training for the overall success flag.

We agree with the reviewer here, but as with the point above, this paper we consider more as a pilot study, where we have tested the viability of this method. We hope future studies will consider this point.

- Section 5: References to Fig. 11 should probably be to Fig. 10. Figure 11 is not discussed at all, as far as I can see.

This was caught by one of the other reviewers as well, references to both Figs 10 and 11 are now clear in section 5.

- Figure 10: Left column repeats the same figure three times, which I guess is a mistake.

These figures had so much data that all of the points were overlayed and thus showing more or less the same thing. We have now split out the pass and failures into two separate columns to give a better idea of the distribution of pass and failure flags. Both Figures 10 and 11 have been updated to reflect this.

- Figure 11: Something is not quite right here – the right figures' colour scale does not seem correct.

As with the point above, apparent error was due to overlapping data points, splitting out the pass and fail flags into the new left and middle columns has helped highlight the correct distribution of flags, and related predicted failures. The updated Figures 10 and 11 answer this point.

- Figure 11 caption: "from a day in 2020" – which day?

Corrected to August 12th.

- L375: "do a good job in predicting the actual failures" – this is not clear from the current set of figures.

We have toned down this statement to "capable of predicting the actual failures"

- L418 and elsewhere: I find the percentage speedups difficult to understand. What is a 100% speedup? At least to me, it would be easier to understand if the reduction in computational time is given.

We understand the point made here, for us we are not sure how beneficial giving exact timings are, as our computational set will differ to those used by other teams, and therefore giving exact figures may not be so useful. What we now provide an additional section of text describing how long our retrievals take, to give a baseline of how we calculate speed ups. This is the new section 2.3.2.

"The TROPESS project has access to computational facilities that includes 100s of individual cores. This processing facility typically allows for the completion of trace gas retrievals in several minutes, with multiple retrievals occurring in parallel. The time for a retrieval depends on the instrument, with AIRS-OMI taking longer than CrIS. Based on the computational facilities available, and the processing times for retrievals, typically a test dataset of around 8000 retrievals takes roughly 2 days to create. Through this paper we refer to how the application of the ML tools allow for a speed-up in processing, this processing benchmark is what we base the speed-up off."

- L439: Again, I'm confused by the speedup given. If 74% of the data is removed, I would either see a speedup by a factor of 4 or a reduction in computational time by 74%.

We decided to remove most references to speed-up in the paper, given that processing set-ups will vary depending on the user, so explicit values are not so useful.

- Figure 15: it is clear from the figure that the filtering is mainly removing cloudy scenes and the right part of the OMI swath

Yes, agreed.

- L459: "This cost/benefit can be improved...". This might be the case, but the authors have not shown any indication of that

We have modified this statement as follows:

"This cost/benefit might be improved with more and sophisticated training of the ML model, potentially to the point where there is very little cost in applying the ML model, which is a topic for further work and exploitation."

- L460: "This work represents the first step in understanding why and how retrievals fail" – I disagree. This is not what this work is about. If you are interested in finding the reasons for failing retrievals, the detailed error information from the OE retrieval will be more helpful.

Our aim with this statement is to identify future work paths, where ML could be used to help refine what causes retrievals to fail. We have modified this statement as follows:

"This work may represent the first step in understanding why and how retrievals fail, for example future work"

- L482: A different name is used here for your ML method than in the description in the text. Please make it consistent.

Corrected to "extremely randomized trees"

- L485: "which can be reduced..." – again, this has not been shown

Changed to "could be reduced"

- L486: "speedup of 66%" – I do not know how you compute these numbers. Only 67% of the original retrievals have to be performed, leading to a reduction of the computational time by 33%. The speedup would be by a factor of 1.47, but as discussed above, I think the computational time is much easier to understand.

As per the related comment above, we have removed direct references to speed-ups, as these specific figures are probably not so useful to other groups. We think references to

how many retrievals are removed from the pipeline are more important than specific speed-up figures.

- Appendix A and B: I do not think that this is needed or adds anything to the manuscript

We have removed Appendix A, however we disagree with the removal of Appendix B (now Appendix A), we think it conveys relevant information about the retrieval windows used in this study.

---

## Author Comment (AC2)

Response to Reviewer 2

Thank you for the review, we respond to your comments below. We have kept your original text in black, our responses are in blue and specific changes are in underlined blue.

Predictions of satellite retrieval failures of air quality using machine learning

Malina et al.

**Summary**

This paper investigates the usefulness of machine learning to streamline data processing for incoming satellite retrievals. They highlight the need for improvements in the computational time to go from level-1 to level-2 data with the ever increasing amount of data being created on a daily basis. The principle they explore in this paper is to use machine learning to remove retrieval failures before the processing stage, therefore reducing the amount of data needing to undergo time-consuming processing. They found the an extremely randomized tree model was the best fit for the task and trained the model on CrIS and AIRS-OMI data for ozone, CO and temperature profile.

Their model performs reasonably well with a few caveats and at a high speed, showing how this could be applied to future EO missions and data processing.

**Major Comments**

This is a well written paper and thorough study that fills an obvious niche. I have a few comments below.

Thank you for these general positive comments.

There is a lot of technical detail in the paper which makes its a long read. Most of it is needed but some sections (e.g. 2.3.1) could be shortened as they're not as relevant to the study.

Thank you for this comment, which is echoed by other reviewers. We have reduced the amount of text in section 2.3.1 and removed Appendix A, as these can be considered superfluous technical detail.

There is a little discussion about the cost/benefit at the end of the paper but there isn't much information/calculations on the actual benefit in terms of computing speeds. It would be good to expand on this point as that is the primary motivation for the paper.

We have cut out major discussion on benefit in computing speed, this is because the computing setup available to the TROPESS project will not be the same as that available to other teams/projects. We have added a new section 2.3.2 that describes in more detail how long retrievals take in the TROPESS/MUSES setup. Based on this information, we indicate through the paper how many retrievals are removed from the pipeline. From this information it is possible to identify how much time could be saved in a serial retrieval, which can then be scaled up for parallel retrieval setups.

I would like to see an expansion in the discussion about what the next steps would be to improve the model and what might be considered a good enough model to be implemented.

We have added the following/modified the following text into the discussion:

"There is a significant cost/benefit aspect to the ML model at this time, where significant processing speed-ups can be achieved, but potentially valuable information may be lost. At this time the ML models are sufficiently developed in order to be deployed in an operational sense, especially with a low threshold value which incurs minimal risk of the loss of valuable retrievals. However, there are clearly more improvement that could be made, for example, the cost/benefit might be improved with more and sophisticated training of the ML model, potentially to the point where there is very little cost in applying the ML model, which is a topic for further work and exploitation. For example, training could be undertaken per region, rather than globally, which may yield improved results. Further, more work can be performed on the QA values that the ML models are trained on. These are currently applied globally, but there could be some value in deriving QA values for distinct regions, and training the ML model on these regions.

As an alternative to regional models, the training data could be carefully constructed to ensure a similar frequency of retrieval failures geographically. Variations across time (night and day, different seasons, cloud coverage etc.) could be balanced in a similar fashion. In terms of ML, the classification performance may be improved by considering more classification methods and particularly more elaborate methods of dimensionality reduction that might be more suitable for spectral data."

**Minor Comments**

Line 28: is this not the case for all of the TROPOMI species, not just ozone?

Yes, correct, we have modified this sentence as follows:

"with many of the TROPOMI retrieval algorithms consisting…"

Line 48-50: These sentences don't scan well

We have changed this paragraph to read as follows.

"In this study, we investigate Machine Learning (ML) methods for predicting failed trace gas retrievals using measured satellite spectra prior to full retrieval. This research builds on previous studies, which focus on pre-selecting or filtering of trace gas retrievals (Mandrake et al., 2013; Mendonca et al., 2021), and offers a global solution."

Line 61: This doesn't scan well, should the 'allowing for multiple different products' be in brackets?

We have changed this paragraph to read as follows:

"The CrIS instrument was chosen for this analysis due to the high data volume and wide spectral range (allowing for multiple different products). CrIS products are currently a key component of TROPESS, where, for example, CrIS ozone retrievals have been used with reanalysis models to understand tropospheric ozone during COVID-19 lockdowns (Miyazaki et al., 2021)."

Line 63: I think these two sentences should be joined.

To us this would be a bit of a mouthful of a sentence. But we have made these two sentences flow together a bit better, given the changes we made above.

Line 124 (and elsewhere): There are some inconsistencies between 'L1b' and 'L1B' throughout the text.

Thank you, we have changed all instances to L1B.

Line 230: Should this be "number of **true** positives"?

Yes, thank you, changed.

Figure 10: The figures on the left appear to be repeated instead of for each species.

Thank you, this point was made by reviewer 1 as well and we repeat the answer here. These figures had so much data that all of the points were overlayed and thus showing more or less the same thing. We have now split out the pass and failures into two separate columns to give a better idea of the distribution of pass and failure flags. Both Figures 10 and 11 have been updated to reflect this.

Line 372: Wording doesn't make sense. Should this be a comma instead of a full stop?

Yes agreed, we have changed to a comma.

Section 5: There appears to be no reference to figure 10 although I think the text is actually meant to be referring to figure 10 but states figure 11. In which case there would be no reference to figure 11.

Good catch, we have changed this text to now reference both figures 10 and 11.

---

## Author Comment (AC3)

Dear Reviewer,

Thank you for the review, we respond to your comments below. We have kept your original text in black, our responses are in blue and specific changes are in underlined blue.

This study employs machine learning (ML) to predict retrieval failures based on measured radiances from sensors (CrIS and AIRS+OMI). By using ML as a pre-processing tool, computational resources can be utilized more effectively, as retrieval algorithms are often computationally intensive due to the high precision and accuracy they require. The manuscript is well-written and provides a good analysis of the ML algorithm's ability to filter spectra based on measured radiances from various instruments.

Thank you for the general positive comments.

I believe the study is suitable for publication after addressing the following minor points:

1. Line 200: Regarding PCA, how much of the variance is explained by the 30 components?

We added the following sentence to provide the requested detail:

"We perform PCA to reduce the number of dimensions to 30, or the dimensionality of the dataset, whichever is lower. 30 principal components account for 98.9%-99.6% of explained variance for the full spectrum, and 99.99% for the fitting spectral regions."

2. Section 4.2: When evaluating feature importance, the conclusion suggests that features outside a given window are as important, if not more so (depending on the retrieved species), as those within the window. If the master quality flag includes information from all windows, this would logically increase the importance of information outside the window. Additionally, the spectrum contains information about $O_3$, CO, and TATM outside their respective windows. Could the ML algorithm be sensitive to these regions as well? If so, could future work explore developing an ML algorithm to extract $O_3$, CO, and TATM directly from the spectra? This can be addressed as potential future work in section 6.

This point is very true, it may be possible to develop a ML algorithm to derive trace gas concentrations directly from measured spectra, as the principle is the same as that shown in this paper. We think it is important to note that, the accuracy of this method remains to be improved, and direct applications of this method to trace gas retrieval

could lead to significant uncertainties that are not traceable due to the black-box nature of ML. We have added a discussion on this point in the paper, as bellow:

"One of the implications of this paper is that ML models can differentiate different atmospheric conditions from measured spectra. This implies that an appropriately trained ML model may be able to infer trace gas concentrations directly from measured spectra, as opposed to using the OEM or other retrieval methods. While this will form future interesting work, the risks of all ML methods, such as appropriate training sets and unintended biases would apply, which would add uncertainties to any retrievals derived from this method."

3. In Section 5, $O_3$ and CO are shown for different ML threshold values. This section could be strengthened by comparing the different filtering thresholds to a truth proxy. This would allow for a clear presentation of how the filtering threshold impacts bias and precision. The current quality filter would be a base line for comparison.

This is a good and valuable point the reviewer makes, and certainly would help strengthen this work. However, we feel that this would be better placed in a follow up paper, that further explores this concept in more depth, after applying additional training methods as described in the discussion section of this paper. Further, this paper already contains extensive technical detail, and we think that adding an additional section as described would make this paper very long.

---

## Author Response (AR2)

Dear Prof. Dr Brunner,

Thank you for your review and assessments, we respond inline below.

Dear authors,

I am pleased to let you know that I accept your manuscript for publication subject to small technical corrections.

The reviewers had asked for minor revisions and I consider your responses to be both careful and appropriate.

Please correct the following small remaining points in the revised manuscript
- Page 2, line 32: What about changing to ".. of any PHYSICS-BASED retrieval algorithm .."?

Agreed, changed.

- Page 3, line 63: The second "TROPESS/CrIS" in this sentence needs to be deleted.

Removed.

- Page 3, line 65: I didn't really understand the meaning of the last part of this sentence ".. and the application of this product"

We have removed this last part as it added little.

- Page 3, line 81: "width" appears twice

Additional removed.

- Page 3, line 85: Change to "... (TROPESS)), offering an opportunity .."

Done.

- Page 4, line 101: add a space before "OMI"

Done.

- Page 10, line 212: Change to ".. probability that THE trace gas retrieval .."

Added.

- Page 33, line 457/458: Remove one of the two appearances of "at this time".

One instance removed.

- Page 33, line 460: "more improvements" instead of "more improvement"

Corrected.

- Page 34, line 488: Change "in the current for" to "in the current form"

Corrected.

In the discussion of the potential usage of the approach for direct trace gas retrievals it might be useful to refer to other existing machine learning/AI based retrieval algorithms such as Van Damme et al. (2017; https://doi.org/10.5194/amt-10-4905-2017).

We have added appropriate references here, to enhance the discussion.

Best regards
Dominik Brunner